# A Blood Biomarker for Duchenne Muscular Dystrophy Shows That Oxidation State of Albumin Correlates with Protein Oxidation and Damage in Mdx Muscle

**DOI:** 10.3390/antiox10081241

**Published:** 2021-08-03

**Authors:** Basma A. Al-Mshhdani, Miranda D. Grounds, Peter G. Arthur, Jessica R. Terrill

**Affiliations:** 1School of Molecular Sciences, The University of Western Australia, 35 Stirling Highway, Perth, WA 6009, Australia; basma.al-mshhdani@research.uwa.edu.au (B.A.A.-M.); peter.arthur@uwa.edu.au (P.G.A.); 2School of Human Sciences, The University of Western Australia, 35 Stirling Highway, Perth, WA 6009, Australia; miranda.grounds@uwa.edu.au

**Keywords:** biomarkers, Cys34 albumin, thiol oxidation, oxidative stress, Duchenne muscular dystrophy, *mdx* mice, ageing, exercise, taurine

## Abstract

Duchenne muscular dystrophy (DMD) is a severe X-linked muscle wasting disease with no cure. While the precise mechanisms of progressive dystropathology remain unclear, oxidative stress caused by excessive generation of oxidants is strongly implicated. Blood biomarkers that could track oxidant levels in tissues would be valuable to measure the effectiveness of clinical treatments for DMD; our research has focused on developing such biomarkers. One target of oxidants that has the potential to be harnessed as a clinical biomarker is the thiol side chain of cysteine 34 (Cys34) of the blood protein albumin. This study using the *mdx* mouse model of DMD shows that in plasma, albumin Cys34 undergoes thiol oxidation and these changes correlate with levels of protein thiol oxidation and damage of the dystrophic muscles. A comparison with the commonly used biomarker protein carbonylation, confirmed that albumin thiol oxidation is the more sensitive plasma biomarker of oxidative stress occurring in muscle tissue. We show that plasma albumin oxidation reflects muscle dystropathology, as increased after exercise and decreased after taurine treatment of *mdx* mice. These data support the use of albumin thiol oxidation as a blood biomarker of dystropathology to assist with advancing clinical development of therapies for DMD.

## 1. Introduction

DMD is a fatal X-chromosome linked disease with an incidence of 1 in 3600–6000 male births (reviewed in [1]). DMD occurs as a consequence of mutations in the dystrophin gene that result in dysfunctional or missing dystrophin protein [2]. An absence of functional dystrophin in skeletal muscles leads to membrane fragility and susceptibility to sarcolemma damage resulting in myofibre necrosis (myonecrosis) and severe loss of muscle mass over time (reviewed in [3,4,5]). While the precise mechanisms of myonecrosis and progressive dystropathology remain unclear, oxidative stress caused by excessive generation of oxidants has been widely implicated (reviewed in [6,7,8,9]).

Oxidants can affect cellular function in a number of ways. Hydroxyl radicals can irreversibly damage macromolecules such as proteins, and cause permanent loss of protein function [10,11]. However, other oxidants such as hydrogen peroxide (H_2_O_2_) and hypochlorous acid (HOCl) can oxidise thiols (RSH) of cysteine residues to form disulfides [12], reviewed in [7]. This is of interest because reversible oxidation of protein thiols has been shown to affect the function of many proteins including protein–protein interactions, signal transduction, regulation of gene expression and protein catabolism [7,13,14,15]. Additionally, unlike the irreversible damage caused by hydroxyl radicals, oxidation to disulfides is biologically reversible.

Studies from our laboratory have investigated the location and consequences of oxidative stress in dystrophic muscle and have shown increased reversible protein thiol oxidation in animal models of muscular dystrophy; this is associated with muscle pathology including myofibre necrosis [8,10,14,16,17,18]. Moreover, drug treatments that decrease protein thiol oxidation are also very effective at reducing dystropathology in the mdx mouse model for DMD [16,19]. These data suggest a causative role of protein thiol oxidation in dystropathology.

There is interest in developing biomarkers that can track oxidative stress in various disease states, especially in biofluids such as blood, since access to human tissues is limited (reviewed in [20,21,22,23]). Popular blood biomarkers, such as the protein carbonyl assay or assays for products of phospholipid oxidation (isoprostanes, malondialdehyde, 4-hydroxynonenal) measure irreversible oxidative damage [24,25]. However, few studies have examined whether changes in such blood biomarkers of oxidative stress reflect changes in levels of oxidative stress and consequent damage in specific tissues [26,27,28,29,30]. Furthermore, no study has examined whether blood biomarkers of oxidative stress reflect changes in the levels of protein thiol oxidation in skeletal muscles. Previous studies have shown changes in protein thiol oxidation occurring in muscle tissue without commensurate changes in protein carbonyls, and MDA [10]. As a consequence, it is unlikely that measures of irreversible oxidation in blood would reflect changes in the levels of protein thiol oxidation in muscle.

Plasma albumin thiol oxidation is a potential biomarker of protein thiol oxidation in muscle tissue. While albumin is a plasma protein, the majority of albumin exists in the interstitium, as it moves from the blood across the capillary wall into the interstitial compartments, and returns to the blood through the lymphatic system [31]. It is consequently exposed to oxidants within tissue and can therefore potentially be used a biomarker of tissue oxidation. The cysteine residue 34 (Cys34) of albumin exists mostly in a reduced state in human plasma, but is susceptible to direct oxidation by oxidants or indirect oxidation via thiol/disulfide (SH/SS) exchange reactions [32,33]. The reduced thiol form of Cys34 represents 70–80% of total albumin in plasma of healthy adults (reviewed in [34,35]). The reversibly oxidized forms of albumin represent about 20–30%, which present as a mixture of disulfides, with low molecular weight thiol compounds such as cysteine, cysteinylglycine, glutathione, homocysteine, and γ-glutamylcysteine [36], and reviewed in [11,37]). In addition, a small fraction (2–5%) of albumin Cys34 exists in the biologically irreversibly oxidised sulfinic (-SO_2_H) and sulfonic acid (-SO_3_H) forms [38,39,40].

Oxidative modifications of serum albumin Cys34 have previously been investigated in various disease states, such as organ failure, kidney diseases, and diabetes mellitus, where increased percentages of reversibly and irreversibly oxidised plasma albumin Cys34 are reported [32,39,41,42,43,44,45] and exercise [46]. However, no previous study has examined whether an increase in the oxidation status of albumin (Cys34) is linked to increased oxidative stress or protein thiol oxidation in specific tissues.

We used the classic mdx mouse model of DMD to evaluate levels of plasma albumin oxidation as a measure of oxidative stress in dystrophic muscles. Young growing mdx mice undergo a period of peak muscle necrosis (myonecrosis) between 21–28 days and, after this acute growth, myonecrosis reduces and stabilises to a relatively low (although variable) level (approximately 6% of each skeletal muscle) by about 8–12 weeks of age [17,47,48,49,50]. In adult mdx mice, exercise is used to increase myofibre damage and myonecrosis with associated increased levels of muscle creatine kinase (CK) in the blood [51,52,53,54] thus enabling potential therapeutic interventions to be more rigorously evaluated in vivo [16,17,49,50,55,56,57]; the effects of different types of exercise on mdx mice and DMD muscles are discussed in a recent review [58]. Few studies have investigated histopathology of mdx mice older than one year, with exception of several papers that report decline in health after one year with loss of body and muscle mass, extensive fibrosis of muscles and reduced lifespan [59,60,61]: from 78 weeks of age (corresponding to 18 months used in our study) the mdx mice had difficulty in obtaining food and water unaided, and they failed to groom themselves [59].

There is increasing interest in identifying and using blood biomarkers to monitor disease progression in mdx mice and DMD boys [62,63,64]. Therefore, in this study we examined the relationship between levels of protein thiol oxidation in muscles, to albumin thiol oxidation in plasma, of dystrophic mdx and control wild type C57Bl10scsn (C57) mice at various ages across their life-span (at 23 days, 6 and 12 weeks and 18 months). These ages were chosen on the basis of changes in muscle pathology, and to include ages that are routinely used experimentally. We also tested mdx mice subjected to a damaging short bout of treadmill exercise (at 6 weeks of age). To compare the sensitivity of albumin thiol oxidation relative to an established biomarker of oxidative changes, we measured protein carbonylation as a marker of irreversible protein oxidation (reviewed in [65]) in muscles and plasma, with correlation analysis performed to determine if plasma levels of albumin thiol oxidation and carbonylation reflect the tissue state. We also compared plasma levels of albumin thiol oxidation and carbonylation with plasma CK (a common blood readout of dystropathology), to see if plasma oxidation correlates with this measure of extent of muscle damage. In addition, to assess the use of plasma Cys34 albumin thiol oxidation as a potential readout of drug efficacy, muscle and plasma thiol oxidation was measured in young mdx mice treated with taurine (from 14 to 23 days). Taurine is an amino acid that protects dystrophic muscles from necrosis and decreases protein thiol oxidation in mdx muscles, as described by many papers from three independent groups [66,67,68,69]. These three experiments in mdx mice demonstrate the response of plasma Cys34 albumin thiol oxidation to changes in dystrophic muscle damage in mdx mice under various conditions.

## 2. Materials and Methods

All chemicals and reagents were purchased from Merck (Melbourne; Victoria; Australia) unless otherwise stated.

### 2.1. Animal Procedures

All experiments were carried out on dystrophic mdx (C57Bl/10ScSnmdx/mdx) and normal wildtype control C57 (C57Bl/10ScSn) mice (the parental strain for mdx) from the Animal Resource Centre, Murdoch, Western Australia. Mice were maintained at the University of Western Australia under standard conditions, with free access to food and drinking water. All animal experiments were conducted in strict accordance with the guidelines of the National Health and Medical Research Council Code of practice for the care and use of animals for scientific purposes (2004), and the Animal Welfare act of Western Australia (2002), and were approved by the Animal Ethics committee at the University of Western Australia.

Mice underwent a single 30 min exercise session on a horizontal rodent treadmill (Columbus Instruments, Columbus, OH, USA), using an established protocol [17]. In brief, the protocol involved a settling (stationary) period for 2 min, an acclimatisation with gentle walking period for 2 min (2 m/min), a warm-up period for 8 min (8 m/min) and the main exercise session of 30 min at a pace of 12 m/min. Mice were sampled immediately following the end of the exercise protocol.

### 2.2. Taurine Treatment

Mdx mice were treated with taurine from 14 days of postnatal age (prior to weaning and the acute onset of myonecrosis that occurs by 21 days), with soft chow containing 4% taurine. Untreated mdx and C57 mice had soft chow without taurine. Each group included pups (*n* = 8) with approximately equal male and female mdx pups, with all males for the C57 group. Mice were sampled at 23 days of age after 9 days of taurine treatment.

### 2.3. Blood and Tissue Collection for Oxidative Stress Analyses

Mice were sampled at 23 days, 6 weeks, 12 weeks or 18 months of age, after cervical dislocation while under terminal anaesthesia (2% *v*/*v* Attane isoflurane, Bomac Animal Health, Hornsby, NSW, Australia). While mice were under terminal anaesthesia, whole blood was collected via cardiac puncture. Immediately, nine parts of blood was added to one part of trapping solution consisting of 62.5 mM methoxypolyethylene glycol maleimide (malpeg, 5000 g/mol, JenKem Technology, Plano, TX, USA) in 40 mM imidazole, pH 7.4 [46]. Blood samples with trapping solution were briefly vortexed and then centrifuged in a refrigerated centrifuged. Plasma was collected before incubating for another 20 min at room temperature before storage at −80 °C until analysis. The remaining blood was immediately centrifuged, plasma removed, and stored at −80 °C until biochemical analysis.

Quadriceps muscles were dissected and frozen immediately in liquid nitrogen and then stored at −80 °C for biochemical analysis.

### 2.4. Protein Thiol Oxidation in Muscle

Reduced and oxidised protein thiols were measured in quadriceps muscles using the two-tag technique as described previously [66]. In brief, frozen muscle was crushed under liquid nitrogen, before protein was extracted with 20% trichloroacetic acid (TCA)/acetone. Protein was solubilized in SDS buffer and protein thiols were labelled with the fluorescent dye BODIPY FL-N-(2-aminoethyl) maleimide (FLM, Invitrogen, Waltham, MA, USA). Following removal of the unbound dye using cysteine, protein was re-solubilized in SDS/Tris (0.5% SDS, 0.5 M Tris, pH 7.0) and oxidized thiols were reduced with tris(2-carboxyethyl)phosphine (TCEP) before the subsequent unlabelled reduced thiols were labelled with a second fluorescent dye Texas Red C2-maleimide (Texas red, Invitrogen). The sample was washed in 100% TCA, followed by acetone, and resuspended in SDS buffer. Samples were read using a fluorescent plate reader (Fluostar Optima) with wavelengths set at excitation 485 nm, emission 520 nm for FLM and excitation 595 nm, emission 610 nm for Texas red. A standard curve for each dye was generated using ovalbumin and results were expressed per mg of protein, quantified using the DC Protein Assay (Bio-Rad, Gladesville, NSW, Australia).

### 2.5. Plasma Albumin Thiol Oxidation

The analysis of plasma thiol oxidation is summarised in Figure 1. Frozen plasma samples containing trapping solution were thawed at 37 °C with agitation before splitting in half to become a reduced (R) sample and a non-reduced control protein sample (NRC). For the NRC aliquot, plasma was diluted (1/99) with SDS/Tris (as above). The R aliquot was diluted in half with 20 mM L-cysteine hydrochloride monohydrate (pH 3), followed by incubation at room temperature for 30 min. The R aliquot was then diluted in half again with 25 mM malpeg before a further incubation of 30 min at room temperature, and then diluted (4 µL in 99 µL) in SDS/Tris [46].

All samples were diluted 1 in 2 with loading buffer containing 0.187 M Tris pH 6.8, 4% SDS, 0.03% *w/v* bromophenol blue, and 30% glycerol. Samples were resolved in 12% acrylamide gels containing 1% (*v*/*v*) of 2,2,2-trichloroethanol for fluorescent (Stain Free, Bio-Rad, Gladesville, NSW, Australia) imaging [70]. Gels were imaged using the Stain-Free imaging program on the ChemiDoc MP Imaging System (Bio-Rad), and loading was checked by measuring albumin signal.

Proteins were transferred to nitrocellulose membranes using the Trans Turbo Blot System (Bio-Rad). Membranes were washed three times with Tris-buffered saline containing Tween 20 (TBST) (10 mM tris, 150 mM NaCl, 0.1% *v*/*v* Tween 20, pH 7). All washing steps were carried out for 5 min with gentle agitation. Membranes were incubated with gentle agitation for 1 h with blocking buffer (5% *w*/*v* skim milk powder in TBST) at room temperature and washed with TBST three times. Immuno-blotting was performed with antibodies to bovine serum albumin (Abcam, 192603, Cambridge, MA, USA) dissolved 1:5000 in TBST. Horseradish peroxidase conjugated goat anti-rabbit secondary antibodies (Thermo Fisher Scientific, Waltham, MA, USA) were diluted 1:10,000 in 5% skim milk in TBST. The ChemiDoc MP Imaging System (Bio-Rad, Gladesville, NSW, Australia) was used to capture Chemiluminescence signal. ImageJ software was used to quantify the resultant images [71]. The background was subtracted, and edited for speckling and noise, before the image was inverted. Signal profile was measured using mean gray value of each band in the membrane. The linearity of signal was checked during method development. For normalisation, a common sample was loaded onto each gel. A quality control sample was used throughout method development, and was run on every gel analysed to ensure reproducibility. The coefficient of variation was 8.1% (*n* = 5). All gels and blots are shown in the Appendix A.

### 2.6. Protein Carbonylation in Muscle and Plasma

Muscle protein carbonyl content was determined by using an immunoassay blot as described previously [72,73] with some modifications. In brief, TCA acetone extractions of quadriceps muscle, were washed with acetone, sonicated, and resuspended in SDS/Tris buffer, before normalisation of protein content using the DC Protein Assay. Muscle supernatants were diluted in extraction buffer to the same protein concentration (1 mg/mL). Two aliquots of protein extracts in SDS/Tris buffer were added into new microfuge tubes and to each 12% SDS was added. To one tube, 10 mM 2,4-Dinitrophenylhydrazine (DNPH) in 10% trifluoroacetic acid was added, followed by a 15 min incubation at room temperature. Carbonyl groups (aldehydes and ketones) react with DNPH to form DNP bound proteins. To the other tube, serving as a negative control, 10% trifluoroacetic acid was added. To all samples, neutralisation/loading solution (2 M Tris, 30% glycerol, 0.02% bromophenol blue) was added, and then samples were loaded onto 4–15% PROTEAN^®^ TGX precast gels (Bio-Rad).

To measure protein carbonyl in plasma, the following protocol was used. Frozen plasma samples were thawed, and then diluted 30 times with ice-cold 1% NP40, 1 mM EDTA in phosphate-buffered saline (PBS), supplemented with complete EDTA free protease inhibitor tablets (Roche, Sydney, Australia). Two aliquots of each plasma sample were diluted 1 in 2 with 6% SDS. To one tube, 10 mM DNPH in 10% trifluoroacetic acid was added to a final DNPH concentration of 5 mM, before incubation for 15 min at room temperature. Following the addition of neutralisation solution, samples were diluted four times with loading buffer (without DTT). Plasma samples were further diluted 2 fold with 2× loading buffer before incubation at 95 °C for 5 min and samples were resolved on 4–15% PROTEAN^®^ TGX gels. A plasma sample from a C57 mouse was incubated with 1 mM HOCl prior to DNPH labelling for use as a positive control.

For quantification of protein carbonyls in both muscle and plasma samples, gels were immunoblotted as described for albumin protein thiol oxidation, except the primary antibody was a rabbit whole antiserum antibody to DNP (D9656, Sigma, St. Louis, MI, USA) diluted 1:20,000 in TBST. Horseradish peroxidase conjugated goat anti-rabbit secondary antibody (Thermo Fisher Scientific) was diluted 1:25,000 in 5% skim milk in TBST. Carbonyl content was calculated as a ratiometric value. In muscle, protein carbonyl was measured as the carbonyl density divided by the amount of fluorescence signal of the whole lane protein content from the stain free gel (arbitrary value). In plasma, albumin carbonyl was calculated as the carbonyl density divided by the amount of fluorescence signal of albumin from the stain free gel. For normalisation, a common sample was loaded onto each gel. All and blots are shown in the Appendix A.

### 2.7. Creatine Kinase (CK) Assay in Plasma

Plasma CK activity reflects the leak of CK from muscles and is a classic measure of damage and necrosis of dystrophic muscles [73]. CK levels were measured in duplicate using the CK-NAC kit (Randox Laboratories, Parramatta, NSW, Australia), and analysed kinetically using a BioTek Powerwave XS Spectrophotometer using the KC4 (V34) program. In brief, plasma was diluted in 0.1% NaCl, before loading into a 96 well plate. Enzyme reagent was added and samples analysed for rate of absorbance change over 30 min, every 1 min, at 340 nm at 37 °C.

### 2.8. Statistics

Significant differences between groups were determined using GraphPad Prism software. Data were analysed using one-way ANOVA tests with post hoc testing, and all data are presented as mean ± standard error of the mean (SEM). Significance was set at *p* < 0.05. Pearson’s correlation was used to assess the relationship between muscle and plasma parameters; all age groups were included in this analysis.

## 3. Results

### 3.1. Protein Thiol Oxidation and Carbonylation in Muscles of C57 and Mdx Mice

The percentage of protein thiol oxidation and carbonylation in muscle (quadriceps) was measured in C57 and mdx mice at 23 days (acute phase of myonecrosis), 6 weeks (initial myonecrosis has reduced and stabilised, approaching adult levels, with and without a single bout of treadmill exercise), 12 weeks (stable adult levels) and 18 months (myonecrosis has reduced further and fibrosis is extensive) (Figure 1). Protein thiol oxidation was higher in 23 day, 6 week and 12 week old mdx muscle compared with age-matched C57 muscle (Figure 2A). Treadmill exercise increased protein thiol oxidation 1.5 fold in 6 week old mdx muscle (Figure 2A). Muscle protein thiol oxidation was also compared between each time point for both strains; in C57 muscle protein thiol oxidation was comparable at 23 day and 6 week, however it increased 1.3 fold by 12 weeks. In mdx muscle, protein thiol oxidation decreased 40% from 23 days to 6 weeks, after which it increased 1.4 fold by 12 weeks. From 12 weeks to 18 months, muscle protein thiol oxidation decreased 17%.

Protein carbonylation was 2.4 fold higher in 23 day old mdx muscle compared with age-matched C57 muscle (Figure 2B). No other difference in muscle protein carbonylation were observed.

### 3.2. Plasma Albumin Thiol Oxidation and Carbonylation in C57 and Mdx Mice

Total albumin thiol oxidation was higher in 23 day, 6 week and 12 week old mdx plasma compared with age-matched C57 plasma (Figure 3A). Treadmill exercise had no effect on total albumin thiol oxidation in 6 week old mdx plasma (Figure 3A). In C57 plasma, total albumin thiol oxidation decreases 46% from 23 day to 6 weeks, however it increases 1.5 fold by 12 weeks (Figure 3A). In mdx plasma, total albumin thiol oxidation decreases 20% from 23 days to 6 weeks (Figure 3A).

There was no difference in reversible albumin thiol oxidation when comparing mdx plasma with age-matched C57 plasma (Figure 3B), and treadmill exercise had no effect on reversible albumin thiol oxidation in 6 week old mdx plasma (Figure 3B). In C57 plasma, reversible albumin thiol oxidation decreases 40% from 23 day to 6 weeks, and it increases 1.6 fold from 12 weeks to 18 months (Figure 3B). In mdx plasma, reversible albumin thiol oxidation decreases 36% from 23 days to 6 weeks (Figure 3B). Irreversible albumin thiol oxidation was 2.8 and 2 fold higher, respectively, in 6 week and 18 month old mdx plasma compared with age-matched C57 plasma (Figure 3C). Treadmill exercise increased irreversible albumin thiol oxidation 1.5 fold in 6 week old mdx plasma (Figure 3C). In C57 plasma, irreversible albumin thiol oxidation decreases 60% from 23 day to 6 weeks, and it increases 2.7 fold from 6 to 12 weeks (Figure 3C). Age had no effect on irreversible albumin thiol oxidation in mdx plasma (Figure 3C).

Protein carbonylation was 2 fold higher in 18 month old mdx plasma compared with age-matched C57 muscle (Figure 3D). In C57 plasma, no differences were observed for protein carbonylation across the age groups, however in mdx mice, protein carbonylation was approximately 2 fold higher in 18 month plasma compared to all other age groups. 

### 3.3. Creatine Kinase (CK) in Plasma of C57 and Mdx Mice

As a measure of mdx muscle dystropathology, plasma CK levels were quantified in all groups. Plasma CK was higher in 23 day, 6 week, 12 week and 18 month old mdx plasma compared with age-matched C57 plasma (Figure 4). Treadmill exercise increased CK levels 5 fold in 6 week old mdx plasma. Age had no effect on CK levels in C57 plasma; in mdx plasma, CK decreased 60% from 23 days to 6 weeks (Figure 4).

### 3.4. Correlation Analysis

To determine if plasma levels of albumin thiol oxidation and carbonylation reflect muscle tissue state, correlation analysis was performed, and we also compared plasma levels of albumin thiol oxidation and carbonylation to levels of plasma CK (the classic readout of dystropathology.

A positive relationship was observed between muscle protein thiol oxidation and total and irreversible plasma albumin thiol oxidation (Table 1) and also between plasma CK and total and irreversible plasma albumin thiol oxidation (Table 1). There was no relationship between muscle carbonylation and plasma CK nor plasma protein carbonylation (Table 1).

### 3.5. Albumin Thiol Oxidation in Plasma of Taurine Treated Mdx Mice

To assess the use of plasma albumin thiol oxidation as a readout of drug efficacy, muscle and plasma thiol oxidation was measured in young mdx mice treated with taurine for 9 days. Firstly, to demonstrate that taurine was effective at decreasing dystropathology in 23 day old mdx mice, plasma CK was measured; taurine treatment decreased CK levels by 30% (Figure 5A). Taurine treatment also decreased protein thiol oxidation by 20% in mdx muscle (Figure 5B) and decreased total albumin thiol oxidation by 17% (Figure 5C) and reversible albumin thiol oxidation by 19% in mdx plasma (Figure 5D). Taurine treatment had no effect on irreversible albumin thiol oxidation in mdx plasma at 23 days.

## 4. Discussion

A key finding of this study is that albumin Cys34 thiol oxidation in plasma correlates with protein thiol oxidation in dystrophic muscle, with both being increased in mdx mice compared with normal C57 mice at multiple time points across the progression of the disease. Our data suggest that plasma albumin oxidation is a more sensitive indicator of oxidative stress in mdx muscle compared with the commonly used biomarker protein carbonylation. We also show that both plasma albumin thiol oxidation and muscle protein thiol oxidation are sensitive to two modulators of dystropathology, treadmill exercise and taurine treatment that, respectively, increase and decrease myonecrosis. Additionally, plasma albumin thiol oxidation correlates with plasma CK levels, a biomarker of dystropathology (myonecrosis).

We measured plasma albumin thiol oxidation at various time points across the lie span of the mdx mice- at 23 days (during the acute phase of myonecrosis), 6 weeks (after the initial phase of myonecrosis has reduced and stabilised, and is approaching adult levels), 12 weeks (stable adult levels of myonecrosis) and 18 months (extensive fibrosis is evident) [47,48,49,50,59]. These ages were chosen on the basis of changes in muscle pathology. The ages also include many (but not all) ages that are used experimentally. We show that increased plasma albumin thiol oxidation occurs in mdx mice at all time points. Furthermore, there were significant correlations between plasma albumin thiol oxidation with protein thiol oxidation in mdx muscle. These data provide the first experimental evidence that plasma thiol oxidation changes reflect the levels of protein oxidation in muscle. Previous research has shown increased oxidation of serum Cys34 albumin in acute-on-chronic liver failure [32], liver cirrhosis [42], chronic renal failure [43], and diabetes mellitus [44], but these studies did not examine whether there were concomitant changes in tissue. We also generated novel data of oxidative state of plasma albumin Cys34 albumin oxidation in healthy mice, total albumin oxidation was between 20 and 40%, reversible was between 15 and 30% and irreversible oxidation was between 6–16%. In comparison, for healthy adult human serum, total Cys34 albumin oxidation is 20–30%, with irreversible oxidation at 2–5% and reversible oxidation at 20–30% (reviewed in [34]).

In adult mdx mice, exercise is routinely used to increase dystrophic myofibre damage [51,52,53,54]; and for this reason we included a study to examine the impact of exercise on albumin thiol oxidation. In the current study, treadmill exercise increased plasma CK and muscle protein thiol oxidation. While no changes in total and reversible albumin thiol oxidation were observed, treadmill exercise of mdx mice increased irreversible albumin thiol oxidation in plasma. Irreversible thiol oxidation includes the formation of sulfinic (-SO_2_H) and sulfonic acid (-SO_3_H) forms [38,39,40], and is generally associated with a loss of function, and degradation of the protein [74].

Our results show that protein carbonylation, a consequence of irreversible oxidative modification of proteins which yields carbonyl groups such as ketones and aldehydes [75] and reviewed in [76,77], occurs primarily on albumin in plasma. This observation is consistent with earlier studies that show plasma albumin is the major carbonylated protein in chronic kidney disease and hemodialysis patients [78,79]. However, we show no strain specific difference in albumin carbonylation when comparing mdx and C57 plasma, apart from at 18 months. In contrast, in mdx muscle protein carbonylation is elevated at 23 days compared to C57 muscle, while there were no difference between mdx and C57 at the other age groups. Moreover, we find no correlation between plasma albumin carbonylation and muscle protein carbonylation. Some animal studies have investigated the relationship between protein carbonylation in plasma and muscle tissue. For example, in rats, protein carbonyls increased in both muscle and plasma after exercise [80]; although another study showed no change in protein carbonylation after acute exercise in rat brain, liver, heart, and skeletal muscle [81]. Moreover, a third study in rats reported that measuring serum protein carbonylation was not a useful indicator of the degree of tissue protein oxidative damage [26]. Taken together, our work, and previous studies indicate that plasma protein carbonylation cannot be assumed to reflect changes in muscle protein oxidation.

To investigate the use of plasma albumin thiol oxidation as a readout dystropathology, young mdx mice were treated with the drug taurine. As per our previous studies, taurine treatment reduced mdx myofibre damage, as measured by plasma CK [66,69,82]. Plasma CK reflects increased sarcolemmal fragility (reviewed in [3,64], which has long been linked to muscle damage in muscular dystrophy [83,84]. In the current study, we show that total and reversible plasma Cys34 albumin thiol oxidation were also decreased in response to taurine treatment of mdx mice, as was muscle protein thiol oxidation. Although irreversible albumin oxidation did not change with taurine treatment, data indicate that plasma total and reversible albumin thiol oxidation reflect both muscle thiol oxidation and damage.

These data also suggest that the mechanism by which taurine may be protecting dystrophic muscle is by decreasing oxidative stress; we have previously proposed that taurine protects dystrophic muscle through its interaction with the neutrophil/HOCl pathway [18]. Neutrophils secrete HOCl, which is a potent ROS that targets proteins by reacting extremely rapidly with thiols and by causing oxidative damage [85]. Taurine can scavenge taurine via the formation of taurine chloramine, a molecule that has additional anti-inflammatory properties (reviewed in [86]). However, further investigation is required to establish the role of HOCl in dystropathology and albumin oxidation.

## 5. Conclusions

In summary, our data show that plasma Cys34 albumin thiol oxidation is a useful blood biomarker of oxidative stress in mdx mice, and that changes in such thiol oxidation in plasma more closely reflect changes in protein thiol oxidation in dystrophic muscle, than the commonly used biomarker protein carbonylation. As discussed in a recent review [64] a range of studies are required to demonstrate that such candidate blood biomarkers for DMD are robust, and relate to the extent of myonecrosis in animal models of DMD, as demonstrated here for in Cys34 albumin in response to exercise, and in taurine treated young mdx mice. In order to further validate the use of Cys34 albumin thiol oxidation as a potential blood biomarker for tracking oxidative stress in DMD, it is desirable to do such measurements in other mammalian models of DMD such as dystrophic rats and dogs and to check the pre-clinical response of this new blood biomarker to beneficial treatments that protect dystrophic muscles from myonecrosis [64], plus to do clinical analyses of blood from a range of DMD patients and control human subjects. The data presented here suggest that albumin Cys34 thiol oxidation has the potential to be a useful robust blood biomarker of dystropathology and may be used to efficiently evaluate and advance the development of therapies for DMD.

## Figures and Tables

**Figure 1 antioxidants-10-01241-f001:**
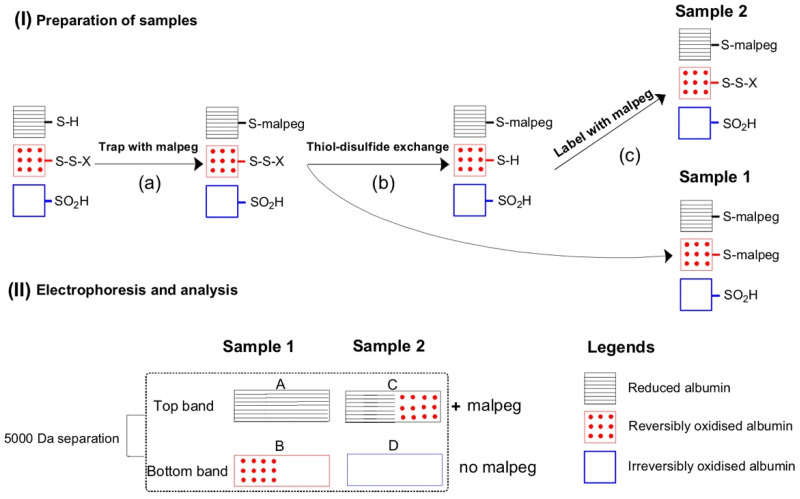
The measurement of irreversibly versus reversibly oxidized albumin Cys 34. (**a**) Represents available thiols (-S-H) in the plasma sample that are initially trapped with malpeg. (**b**) The sample is divided in two, with reversibly oxidised thiols (S-S-X) in the second sample converted to reduced thiols by thiol-disulphide exchange reactions and then (**c**) the reduced thiols are labelled with malpeg. Following electrophoresis, albumin bound to malpeg is separated by 5000 Da figure.

**Figure 2 antioxidants-10-01241-f002:**
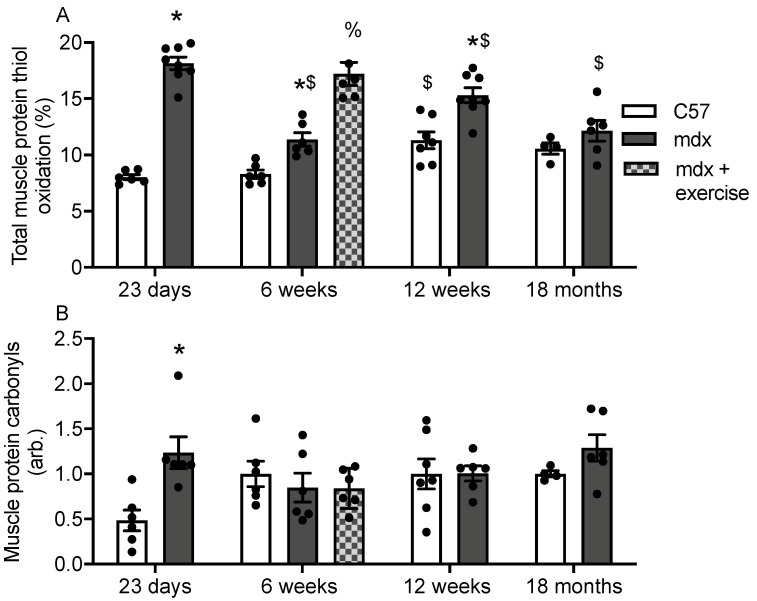
Protein thiol oxidation (**A**) and protein carbonylation (**B**) in C57 and mdx muscle at various time points. * = significantly (*p* < 0.05) different to age matched control. % = significant (*p* < 0.05) different to unexercised 6 week old mdx. $ = significantly (*p* < 0.05) different to previous age group of same strain. Bars represent mean ± SEM and *n* = 5–8 per group.

**Figure 3 antioxidants-10-01241-f003:**
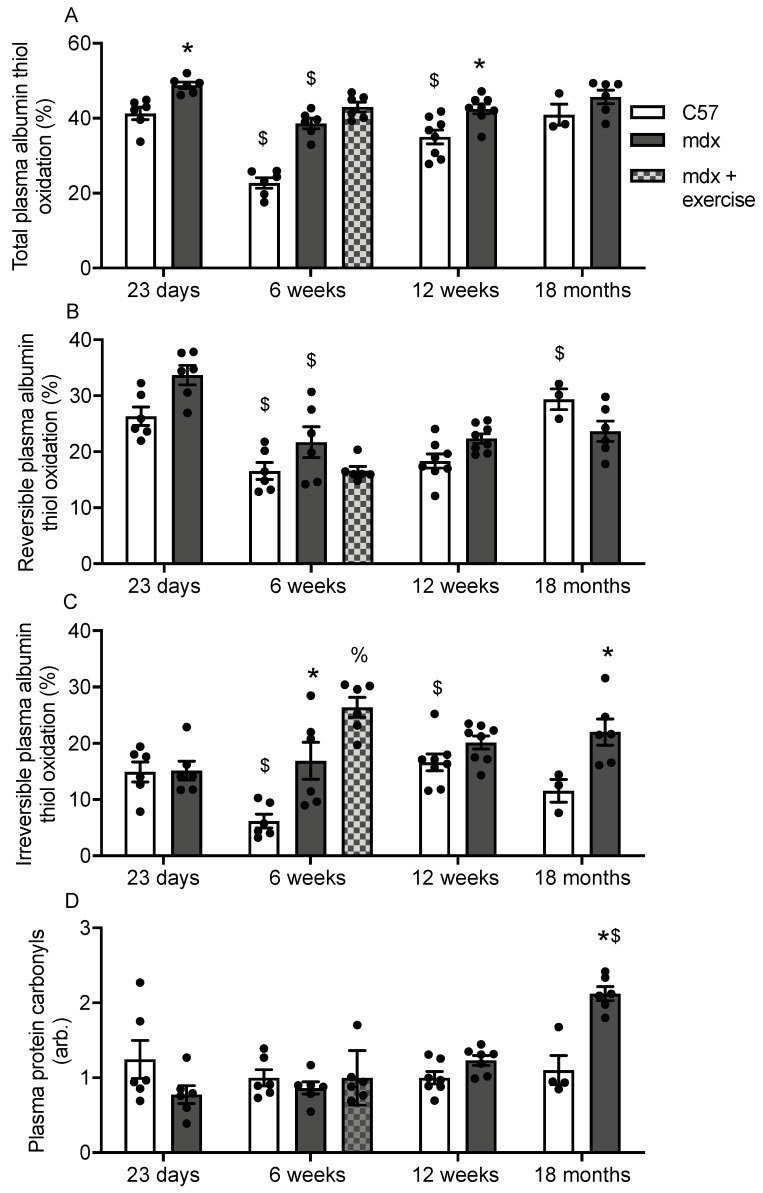
Total albumin thiol oxidation (**A**), reversible albumin thiol oxidation (**B**), irreversible albumin thiol oxidation (**C**) and protein carbonylation (**D**) in C57 and mdx plasma at various time points. * = significantly (*p* < 0.05) different to age matched control. % = significant (*p* < 0.05) different to unexercised 6 week old mdx. $ = significantly (*p* < 0.05) different to previous age group of same strain. Bars represent mean ± SEM and *n* = 5–8 per group.

**Figure 4 antioxidants-10-01241-f004:**
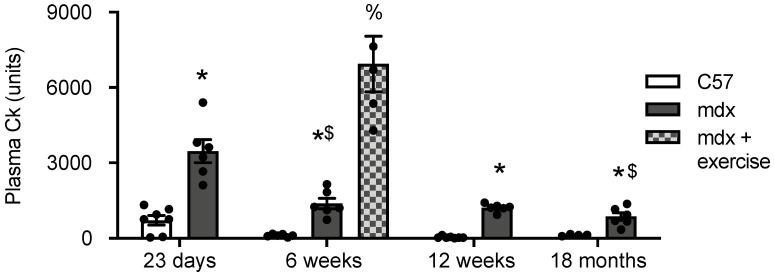
Creatine Kinase (CK) in C57 and mdx plasma at various time points. * = significantly (*p* < 0.05) different to age matched control. % = significant (*p* < 0.05) different to unexercised 6 week old mdx. $ = significantly (*p* < 0.05) different to previous age group of same strain. Bars represent mean ± SEM and *n* = 5–8 per group.

**Figure 5 antioxidants-10-01241-f005:**
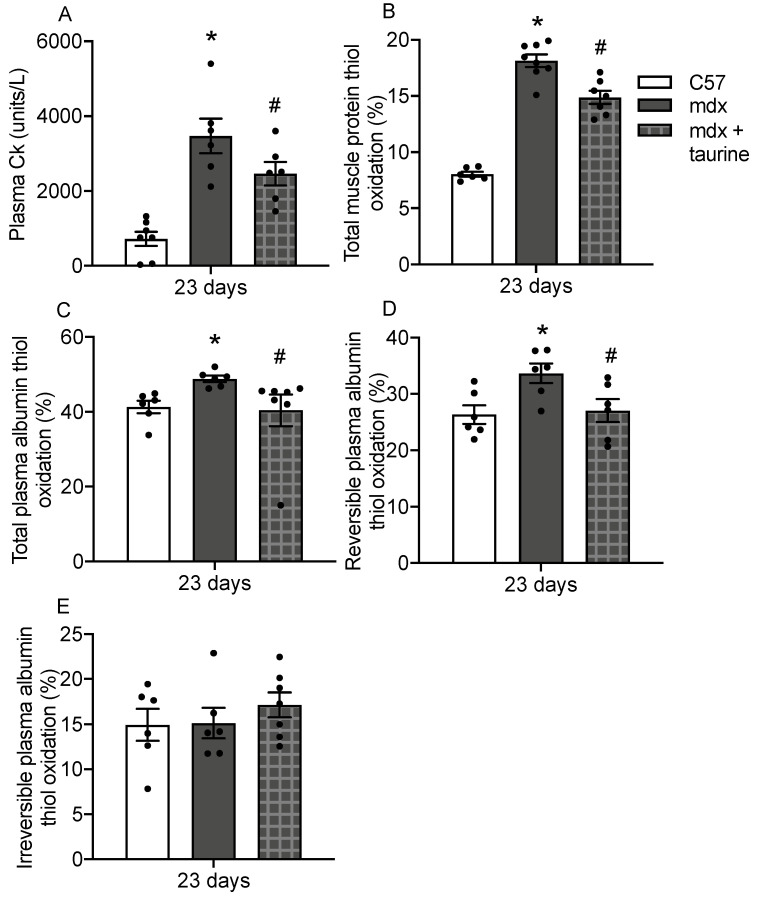
Plasma CK (**A**), total muscle protein thiol oxidation (**B**), total plasma albumin thiol oxidation (**C**), reversible plasma albumin thiol oxidation (**D**), irreversible plasma albumin thiol oxidation (**E**) and protein carbonylation (**D**) in C57 and mdx plasma at various time points. * = significantly (*p* < 0.05) different to age matched control. # = significant (*p* < 0.05) different to untreated mdx. Bars represent mean ± SEM and *n* = 5–8 per group.

**Table 1 antioxidants-10-01241-t001:** Correlation between muscle and plasma protein thiol oxidation (PTO) and protein carbonylation (prot. carb.) makers, and between plasma CK and plasma protein oxidation markers in C57 and mdx mice. *N* = 54 and asterisks represent significant correlation of *p* < 0.05.

Indices 1	Indices 1	*r*	*p*
Muscle PTO	Plasma total PTO	0.5	<0.0001 *
Muscle PTO	Plasma revers. PTO	0.15	0.3303
Muscle PTO	Plasma irrevers. PTO	0.5	0.0002 *
Muscle prot. carb.	Plasma prot. carb.	0.05	0.7
Plasma CK	Plasma total PTO	0.4	0.007 *
Plasma CK	Plasma revers. PTO	0.04	0.8
Plasma CK	Plasma irrevers. PTO	0.4	0.004 *
Plasma CK	Plasma prot. carb.	-0.1	0.3

## Data Availability

Data is contained within the article and Appendix A.

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
