# Peer review of "A Blood Biomarker for Duchenne Muscular Dystrophy Shows That Oxidation State of Albumin Correlates with Protein Oxidation and Damage in Mdx Muscle"

_antioxidants, 2021, doi:10.3390/antiox10081241_

Round 1
Reviewer 1 Report
Comment on the manuscript entitled “A promising blood biomarker for Duchenne muscular dystro-2 phy shows that the oxidation state of plasma albumin corre-3 lates with protein oxidation and damage in dystrophic mdx 4 muscle 5” by Al-Mshhdani et al.
Significance and impact?
This study describes the use of plasma cys34 albumin thiol oxidation as a biomarker to assess levels of oxidative stress in the mdx mouse model for Duchenne muscular dystrophy (DMD). The need of reliable and sensitive biomarker to assess DMD disease progression and response to therapies is highly important for drug development program. The data shows a correlation between the plasma level of albumin thiol oxidation and muscle protein carbonylation, providing a link between the oxidative state of muscle and circulating albumin. The authors also claim that Cys34 correlated with CK levels and responded to taurine treatment. This is an interesting finding, and this could contribute immensely to the field of DMD and outcome measures. However, there are major concerns that need to be addressed by the authors to strengthen this study.
(1) The authors based their measurement of western blots to ass oxidative stress in muscle and to measure levels of oxidized and non-oxidized Cys34 in Albumin. Description of the reproducibility and precision of the method is lacking. How exactly the authors measured irreversibly (-SO2H) and -SO3H) versus reversibly oxidized Cys (S-S) at position 34 is not clear. No western blot images with loading controls are shown. What loading control was used for data normalization. This is highly important for assay precision and could introduce technical variability if not correctly done.
(2) While the research design and methods are thoroughly explained, a QC to ensure accuracy and precision of the quantitative analysis is lacking. How many times a same sample was measured and what was the CV?
(3) There was an assumption that the decrease in albumin thiol oxidation by taurine indicate a readout of drug efficacy. However, the measurement of irreversible oxidized albumin was not significant between taurine treated and untreated group. The authors should comment on this point.
The overall idea of using oxidized Cys34 in albumin as a tool to assess dystropathology in DMD is attractive and could be useful for to assess disease progression and response to therapies in DMD. However, there are several concerns that dampens the present study. These includes the specificity and sensitivity of the targeted biomarker, the method used to measure it as well as the lack of assessment of other factors involved in the regulation of oxidative stress in the body such plasma free thiols (e.g. methionine) and glutathione levels that could influence the levels of oxidized albumin.
Minor comments
In method section page 4 the authors mention the following
2.4. Protein thiol oxidation in muscle. Under this paragraph the authors refer to procedure reference 65 then in brie they describe how to measure total protein concentration rather than a brief description of how thiol oxidation measurement using a two-tag technique works
Author Response
(1) The authors based their measurement of western blots to ass oxidative stress in muscle and to measure levels of oxidized and non-oxidized Cys34 in Albumin. Description of the reproducibility and precision of the method is lacking. How exactly the authors measured irreversibly (-SO2H) and -SO3H) versus reversibly oxidized Cys (S-S) at position 34 is not clear. No western blot images with loading controls are shown. What loading control was used for data normalization. This is highly important for assay precision and could introduce technical variability if not correctly done.
Response:
Quality controls were used throughout method development and analysis to ensure reproducibility; this has now been mentioned in the Methods (lines 203-204). The measurement of irreversibly (-SO2H) and -SO3H) versus reversibly oxidized Cys has been more clearly explained by the addition of a diagram in the Methods section. We did not include western blot images in the actual manuscript as the instructions to authors suggest that these be included in the supplementary files. We used a fluorescent protein stain to directly assess the loading of albumin and this is now mentioned in the methods (lines 185-187).
(2) While the research design and methods are thoroughly explained, a QC to ensure accuracy and precision of the quantitative analysis is lacking. How many times a same sample was measured and what was the CV?
Response: Please see response to (1.1) above. In addition, precision of the quantitative analysis has now been included (CV was 8.1% [n=5]) in the methods (line 204).
(3) There was an assumption that the decrease in albumin thiol oxidation by taurine indicate a readout of drug efficacy. However, the measurement of irreversible oxidized albumin was not significant between taurine treated and untreated group. The authors should comment on this point.
Response: This has now been addressed in the discussion, line 423.
(4) The overall idea of using oxidized Cys34 in albumin as a tool to assess dystropathology in DMD is attractive and could be useful for to assess disease progression and response to therapies in DMD. However, there are several concerns that dampens the present study. These includes the specificity and sensitivity of the targeted biomarker, the method used to measure it as well as the lack of assessment of other factors involved in the regulation of oxidative stress in the body such plasma free thiols (e.g. methionine) and glutathione levels that could influence the levels of oxidized albumin.
Response:
In DMD boys, a reliable biomarker of oxidative stress in blood has the potential to provide more insight into the role of oxidative stress in the course of the disease and may also be useful for tracking the effectiveness of clinical treatments for DMD. For a blood biomarker to be useful in DMD, changes in blood need to reflect changes in muscle. This was our objective in this study, and realistically, it could only be carried out in an animal model of dystrophy because of the limited possibility of obtaining muscle tissue from DMD boys. This study established that albumin thiol oxidation in blood was reflective of changes in muscle protein thiol oxidation, and that it was more sensitive than the commonly used protein carbonyl assay. We understand that further work, as indicated in the discussion, is required to establish whether Cys34 albumin thiol oxidation will be useful blood biomarker for tracking oxidative stress in DMD. However, we consider studies to address sensitivity and specificity are better addressed in human subjects because the data would be more relevant.
The issue of specificity is not restricted to albumin thiol oxidation, but to all oxidative stress measures in general. Other oxidative stress measures have been suggested as being useful in understanding and tracking the course of the disease (see, for example: Greilberger J, et al., Malondialdehyde, carbonyl proteins and albumin-disulphide as useful oxidative markers in mild cognitive impairment and Alzheimer's disease. Free Radic Res, 2008. 42:633-8).
We agree with the reviewer that many factors could be affecting the levels of oxidised albumin in plasma. However, our question was whether dystrophy which causes substantial oxidative stress and inflammation in the muscle would also cause an increase in plasma albumin oxidation. Our findings support previous suggestions that albumin thiol oxidation may be a use global biomarker in various human diseases (see, for example: Colombo G, Clerici M, Giustarini D, Rossi R, Milzani A, Dalle-Donne I, Redox albuminomics: oxidized albumin in human diseases. Antioxid Redox Signal, 2012. 17:1515-27.)
Minor comments
In method section page 4 the authors mention the following
2.4. Protein thiol oxidation in muscle. Under this paragraph the authors refer to procedure reference 65 then in brie they describe how to measure total protein concentration rather than a brief description of how thiol oxidation measurement using a two-tag technique works
Response: This was an oversight and has now been corrected to include the whole method on page 4.
Reviewer 2 Report
In this manuscript the authors report that albumin thiol oxidation correlates with protein thiol oxidation in dystrophic skeletal muscle, using the traditional mdx DMD mouse model. While the topic might be somewhat interesting, the data presented here are superfacial and the manuscript lacks mechanistic insights that advance the knowledge on muscle dystropathology. I have the below major concerns:
(1) Please further elaborate on how muscle pathology results in blood albumin thiol oxidation. What is the molecular mechanism for this process?
(2) Validation of the findings in other muscular dystrophy mouse model/human patients/murine muscle injury models (such as cardiotoxin or BaCl2 injury) should be included to demonstrate the reproducibility of the data and significance of the work.
(3) Can authors examine protein thiol oxidation using independent experimental approaches?
(4) Histological analysis of the muscle samples used in the study should be included.
(5) The title is too long. Please be concise and revise the title.
Author Response
(1) Please further elaborate on how muscle pathology results in blood albumin thiol oxidation. What is the molecular mechanism for this process?
Response: While albumin is a plasma protein, the majority of albumin exists in the interstitial fluid that is in intimate contact with cells, as it moves from the blood across the capillary wall into the interstitial compartment, and returns to the blood through the lymphatic system. It is consequently in close proximity to oxidants within cells of tissues and can therefore potentially be used a biomarker of tissue oxidation. For skeletal muscle, it is well documented that albumin can enter into myofibres where the sarcolemma becomes even slightly ‘leaky’, although the basement membrane is intact, demonstrating that the albumin is present in very close proximity to the myofibre surface. This information has now been included in the introduction (lines 66-70).
(2) Validation of the findings in other muscular dystrophy mouse model/human patients/murine muscle injury models (such as cardiotoxin or BaCl2 injury) should be included to demonstrate the reproducibility of the data and significance of the work.
Response: This paper was focussed on Duchenne muscular dystrophy (DMD) only; however we have also validated this method in exercise (see reference Lim, 2020 referenced in the article), and we have papers in preparation that examine it in conditions such as chronic fatigue syndrome. Clearly many future experiments can be done to explore this topic further in a wide range of other dystrophies, neuromuscular disorders and other experimental situations, and we are currently undertaking collaboration with researchers to do this in other conditions. This represent a large amount of future work that falls outside the scope of the present paper and cannot be undertaken at this stage.
(3) Can authors examine protein thiol oxidation using independent experimental approaches?
Response: As above (2.2), our observation in this study includes 3 experimental situations for DMD, plus were have already engaged in a series of other studies and these all form the basis for many other potential future studies.
(4) Histological analysis of the muscle samples used in the study should be included.
Response: This is not possible, since muscles were not specifically collected for histology (this involves either careful freezing in isopentane to preserve tissue morphology, or fixation for subsequent paraffin processing, before cutting and staining sections for histological analysis).
(5) The title is too long. Please be concise and revise the title.,
Response: Title has been changed to “A blood biomarker for Duchenne muscular dystrophy shows that the oxidation state of albumin correlates with protein oxidation and damage in mdx muscles”.
Reviewer 3 Report
The authors Al-Mshhdani et al., made an effort in seeking a blood biomarker for DMD. The authors mainly focused on albumin thiol oxidation and protein carbonylation content in the plasma. The protein carbonylation was reported in the previous findings to not be an appropriate biomarker for DMD which was also confirmed in this manuscript. Instead, the authors find albumin thiol oxidation to be an more relative biomarker for DMD. The concept did make an advance in terms of monitoring disease progression. However, there are several downfalls in terms the design of the experiments.
- The age of the animal being tested were 23 days, 6 weeks, 12 weeks and 18 months. It seems the mice were either young or old. There perhaps should be a group that is 10- to 12-month old. Otherwise, please provide justification.
- In all the experiments, the treadmill exercise is only applied to the 6-week group which is not a standard practice. At 12 week and 18month, there should also be the treadmill exercise group.
- There is no visualized presentation for mdx disease progression in correlation with the CK and oxidation analysis. Perhaps, some muscle cross section can be presented here as the correlation for the pathology of dmd.
- The authors only measured oxidized protein of muscle in quadriceps. What about other muscles? For example, diaphragm.
- In table 1, what is the age of mice when the correlation was measured? Or are the correlation include all the combined data from different age groups?
- In result 3.5, the authors used taurine as the established therapy for DMD. The authors did not provide any citations to validate the effectiveness of taurine. Otherwise, the manuscript should include the evaluation of taurine, not only with CK measurement. Also, authors only measured the data on day 23. Another timepoint should be included.
Author Response
(1) The age of the animal being tested were 23 days, 6 weeks, 12 weeks and 18 months. It seems the mice were either young or old. There perhaps should be a group that is 10- to 12-month old. Otherwise, please provide justification.
Response: This study is comprehensive with analyses at 4 key ages already (for reasons detailed in the paper), including older mdx mice aged 18 months that are rarely used in mdx studies. We do not think there is particular merit in looking at 10-12 months old mdx mice, as these should be similar to the relatively stable disease classically studied in adult mdx mice aged 3 months. We do not feel there is any strong justification to include such an intermediate extra group of mdx mice aged 10-12 months.
Furthermore, all samples from the mdx mice at the 4 ages in this study were analysed together at the same time, so that data are directly comparable. However, an additional group of mice aged 10-12 months would have to be analysed as a separate group, not directly comparable with these existing data. Finally, ageing of additional cohorts of WT and mdx mice up to 10-12 months would incur a long delay of at least a year and is very expensive to do: we do not have the resources and time to do this and so is not feasible.
(2) In all the experiments, the treadmill exercise is only applied to the 6-week group which is not a standard practice. At 12 week and 18month, there should also be the treadmill exercise group.
Response: We disagree that treadmill exercise in 6-week-old is not standard practice, as we have performed it many times, as have other groups (ref 53 for example), with a similar response of increased necrosis of dystrophic mdx muscles to exercise widely reported at different ages. We observe a similar response in 6 and 12-week-old mice. This was a specific experiment to see if exercise affected thiol oxidation, and it did: we do not feel that additional ages would be of any further benefit .
(3) There is no visualized presentation for mdx disease progression in correlation with the CK and oxidation analysis. Perhaps, some muscle cross section can be presented here s the correlation for the pathology of dmd.
Response: The histology of mdx muscle, and the correlation with CK levels at different ages has been described extensively in previous research papers, including in exercised mice (see many references in the paper, e.g – refs.16, 17, 59-61).
(4) The authors only measured oxidized protein of muscle in quadriceps. What about other muscles? For example, diaphragm.
Response: There are a multitude of muscles that could be analysed and agree that many others including diaphragm are also of interest. However, we sampled only quadriceps for this study as it is a large muscle group and the histopathology is well documented, plus we know that the quadriceps responds to exercise by increased myonecrosis. We did not sample the diaphragm, and this is less likely to be so directly impacted by the exercise intervention. A wide range of additional muscles could be of interest to sample and analyse in future studies.
(5) In table 1, what is the age of mice when the correlation was measured? Or are the correlation include all the combined data from different age groups?
Response: All ages groups were included in this analysis, this is now specified in the text (line 263).
(6) In result 3.5, the authors used taurine as the established therapy for DMD. The authors did not provide any citations to validate the effectiveness of taurine. Otherwise, the manuscript should include the evaluation of taurine, not only with CK measurement. Also, authors only measured the data on day 23. Another timepoint should be included.
Response: We did cite many studies in the introduction validating the effectiveness of taurine, but we have now expanded the discussion of it (lines 116-118).
We do not feel that another time point will add any additional information, since we and others have observed a similar beneficial response to taurine in mdx muscles at various ages in prior studies, such 23 days, 6, 4 and 8 weeks and 2.5 months (refs 66, 69, 53. & 82).
Round 2
Reviewer 1 Report
The authors responded to all the questions raised by the reviewers and revisions were made accordingly.
Author Response
All concerns addressed.
Reviewer 2 Report
In the revised manuscript the authors properly addressed my concerns #1 and #5.
For comment #2, I respect authors' research plan on what to include and what not to include for a manuscript. However, I do need to point out that the mdx mouse model is not the best model for DMD available right now, as the nonsense mutation in exon 23 of the mdx mice is not a human mutation, and mdx mouse model does not recapitulate many human DMD phenotypes. Since the authors refused to provide validation using other independent muscle disease/injury models (even the easiest BaCl2 injury model which only involves WT mice, does not need to involve the same aging or exercise protocol), whether the discoveries reported in this paper is solid or an artifact due to the background of a particular mouse model cannot be evaluated and thus is questionable.
For comment #3, the authors only used one experimental method to examine thiol oxidation level. Without validation from another independent method, it's hard to justify the scientific soundness of the approach.
For comment #4, It's regrettable that it is impossible for authors to include histology if they did not collect tissue for this at the first beginning. The experiments could have been better designed.
In summary, the scientific conclusion of this paper is based on the readout of one single methodology of one single genetic mouse model (at different age/exercise condition, which did not add much). Thus, it does not meet the rigor and standard for publication in Antioxidants.
Author Response
- In the revised manuscript the authors properly addressed my concerns #1 and #5.
For comment #2, I respect authors' research plan on what to include and what not to include for a manuscript. However, I do need to point out that the mdx mouse model is not the best model for DMD available right now, as the nonsense mutation in exon 23 of the mdx mice is not a human mutation, and mdx mouse model does not recapitulate many human DMD phenotypes.
It is widely recognised that there are differences between severity of manifestation of muscular dystrophies in different species, due to the very small size, brief growth phase and short life span of mice, compared with humans which we have reviewed (See Grounds 2008 below). We recognise that there are limitations to the use of the mdx mouse model of DMD, but there is wide and continuing literature in support of the mdx mouse, including variations on different background strains and with other mutations (e.g. see van Putten et al, 2020 below). In this context, the mouse model of DMD was suitable to address the specific question we asked: could oxidation cys34 of albumin be used as a biomarker of protein oxidation in dystrophic mdx muscle tissue? As we discussed in the introduction (lines 56-58) an important but often overlooked aspect towards validating a blood biomarker is to establish that the changes in the blood are reflective of changes in the muscle.
With our novel finding that there is a relationship, we can now extend our studies to other animal models of dystrophy, such as the challenging dystrophic dog model. We have previously published with the dog model (see Terrill et al, 2016 below). There is limited access to plasma (which has to be trapped at the time of collection) and tissue in dystrophic dogs, so the current study will provide evidence to undertake further testing the usefulness of cys34 oxidation as a biomarker of protein oxidation in muscle tissue.
- Grounds, MD (2008) Two-tiered hypotheses for Duchenne muscular dystrophy. Cell Mol Life Sci. 65(11) 1621-1625
- van Putten M, Lloyd E, de Greef J, Raz V, Willmann R, Grounds MD (2020) Mouse models for muscular dystrophies: an overview. Disease Models and Mechanisms. 13(2)
- Terrill JR, Duong MN, Turner R, Le Guiner C, Boyatzis, A, Kettle AJ, Grounds MD, Arthur PG. Levels of inflammation and oxidative stress, and a role for taurine in dystropathology of the Golden Retriever Muscular Dystrophy dog model for Duchenne Muscular Dystrophy. Redox Biology. 9 276-286.
- Since the authors refused to provide validation using other independent muscle disease/injury models (even the easiest BaCl2 injury model which only involves WT mice, does not need to involve the same aging or exercise protocol), whether the discoveries reported in this paper is solid or an artifact due to the background of a particular mouse model cannot be evaluated and thus is questionable.
We have extensive experience (see first four refs below) in dystrophy research with many models of experimental muscle necrosis and regeneration, and each model has its own merits. The suggested use of BaCl2 for chemical injury is not a widely used model in dystrophy research, and can produce heterogeneous muscle damage (as shown by study that carefully compared 4 models of experimental injury; see Hardy et al., 2016 below). We do not see how the BaCl2 injury model, which would require several months of work (including ethics applications), would provide sufficient additional relevant information with respect to muscular dystrophy. With such additional work we wold also not be able to meet the deadline for the special issue of 31 August 2021.
The reviewer is concerned that we have only measured protein thiol oxidation in muscle of mdx mice. However, we have undertaken extensive studies investigating protein thiol oxidation. We have shown the protein thiol oxidation is elevated in different muscles from mdx mice (El-Shafey 2011 below), is elevated in muscle of dystrophic dogs (Terrill 2016 below) and found that protein thiol oxidation is also elevated in muscle from dystrophic rats (manuscript in preparation). Furthermore, we have shown that protein thiol oxidation is responsive to N-acetylcysteine (a thiol antioxidant, Terrill 2012 below). In the submitted manuscript we did use taurine to show that protein thiol oxidation was responsive to decreases myonecrosis and muscle inflammation. We also undertaken work in humans and shown that that protein thiol oxidation is responsive to exercise (Lim 2020 below).
- McGeachie JK, Grounds MD, Partridge TA, Morgan JE (1993) Age-related changes in replication of myogenic cells in mdx mice: quantitative autoradiographic studies. Neurol. Sci. 119 169-179.
- Grounds MD, Torrisi J (2004) Anti-TNF alpha (Remicade (R)) therapy protects dystrophic skeletal muscle from necrosis. FASEB J. 18 676-682.
- Radley HG, Grounds, MD (2006) Cromolyn administration (to block mast cell degranulation) reduces necrosis of dystrophic muscle in mdx mice. Dis. 23, 387-397.
- Radley-Crabb HG, Terrill JR, Shavlakadze T, Tonkin J, Arthur PG, Grounds MD (2012) A single 30 min treadmill exercise session is suitable for ‘proof-of concept studies’ in adult mdx mice: A comparison of the early consequences of two different treadmill protocols. Disord. 22 170-182.
- Hardy et al (2016) Comparative Study of Injury Models for Studying Muscle Regeneration in Mice PLoS One. 2016; 11(1): e0147198.
- El-Shafey AF, Armstrong AE, Terrill JR, Grounds MD, Arthur PG (2011) Screening for increased protein thiol oxidation in oxidatively stressed muscle tissue. Free Radic. Res. 45 991-999.
- Terrill JR, Radley-Crabb HG, Grounds MD, Arthur PG, (2012) N-Acetylcysteine treatment of dystrophic mdx mice results in protein thiol modifications and inhibition of exercise induced myofibre necrosis. Disord. 22 427-34
- Lim ZX, Duong MN, Boyatzis AE, Golden E, Vrielink A, Fournier PA, Arthur PG (2020) Oxidation of cysteine 34 of plasma albumin as a biomarker of oxidative stress. Free Radic Res 54(1) 91-103.
- For comment #3, the authors only used one experimental method to examine thiol oxidation level. Without validation from another independent method, it's hard to justify the scientific soundness of the approach.
We are very confident of our method of measuring protein thiol oxidation and have undertaken extensive validation of the method and have used it many previous publications (see below). Measuring protein thiol oxidation in tissue is challenging because protein thiols are labile and are very susceptible to artefactual oxidation during sample preparation. We developed and validated the method for measuring protein thiols in cells and tissue and have compared it with other common measures of oxidative stress (Lui, Armstrong, and El-Shafey below). We have established that protein thiol oxidation is responsive to an external oxidative stress in cultured cells (Tan 2015 below). With respect to albumin thiol oxidation method, we have previously compared the method with two methods: an established chromatography technique and a method that used mass spectrometry to measure oxidation at the residue level (Lim 2020 below).
- Lui JK, Lipscombe R, Arthur PG (2010) Detecting Changes in the Thiol Redox State of Proteins Following a Decrease in Oxygen Concentration Using a Dual Labeling Technique. Proteome Res. 9(1): 383-392.
- Armstrong AE, Zerbes R, Fournier PA, Arthur PG (2011) A fluorescent dual labeling technique for the quantitative measurement of reduced and oxidized protein thiols in tissue samples. Free Radic Biol Med50(4) 510-517
- El-Shafey AF, Armstrong AE, Terrill JR, Grounds MD, Arthur PG (2011) Screening for increased protein thiol oxidation in oxidatively stressed muscle tissue. Free Radic. Res. 45 991-999.
- Tan PL, Shavlakadze T, Grounds MD and Arthur PG (2015) Differential thiol oxidation of the signaling proteins Akt, PTEN or PP2A determines whether Akt phosphorylation is enhanced or inhibited by oxidative stress in C2C12 myotubes derived from skeletal muscle. Int J Biochem Cell Biol 62 72-79.
- Lim ZX, Duong MN, Boyatzis AE, Golden E, Vrielink A, Fournier PA, Arthur PG (2020) Oxidation of cysteine 34 of plasma albumin as a biomarker of oxidative stress. Free Radic Res 54(1) 91-103.
- For comment #4, It's regrettable that it is impossible for authors to include histology if they did not collect tissue for this at the first beginning. The experiments could have been better designed.
We have previously characterised histopathology in mdx muscles in many studies (see refs in query 2 above). These events and incidence of necrosis are highly reproducible and well described. In previous publications we have also related protein thiol oxidation to histopathological changes (see Iwasaki, Terrill, and Radley-Crabb references below). We are therefore confident of the relationship between histology and protein thiol oxidation.
- Iwasaki T, Terrill JR, Shavlakadze T, MGrounds MD, Arthur PG (2013) Visualizing and quantifying oxidized protein thiols in tissue sections: a comparison of dystrophic mdx and normal skeletal mouse muscles. Free Radic Biol Med 65 1408-1416.
- Terrill JR, Webb SM, Arthur PG, Hackett MJ (2020) Investigation of the effect of taurine supplementation on muscle taurine content in the mdx mouse model of Duchenne muscular dystrophy using chemically specific synchrotron imaging. Analyst 145(22) 7242-7251.
- Radley-Crabb HG, Terrill JR, Shavlakadze T, Tonkin J, Arthur PG, Grounds MD (2012) A single 30 min treadmill exercise session is suitable for ‘proof-of concept studies’ in adult mdx mice: A comparison of the early consequences of two different treadmill protocols. Disord. 22 170-182.
- Terrill JR, Radley-Crabb HG, Grounds MD, Arthur PG, (2012) N-Acetylcysteine treatment of dystrophic mdx mice results in protein thiol modifications and inhibition of exercise induced myofibre necrosis. Disord. 22 427-34
- Terrill JR, Radley-Crabb HG, Iwasaki T, Lemckert FA, Grounds MD, Arthur PG (2013) Oxidative stress and pathology in muscular dystrophies: focus on protein thiol oxidation and dysferlinopathies. FEBS J. 2804149-4164.
- In summary, the scientific conclusion of this paper is based on the readout of one single methodology of one single genetic mouse model (at different age/exercise condition, which did not add much). Thus, it does not meet the rigor and standard for publication in Antioxidants.
We disagree with the negative conclusions of this reviewer. The reviewer may not be aware that we have extensively published on the methodology and the genetic model (see previous citations). This manuscript is not stand alone: the findings of this submission extend observations from the previous publications. Furthermore, the manuscript addresses the question that was asked: could oxidation cys34 of albumin be used as a biomarker of protein oxidation in dystrophic mdx muscle tissue? As we discussed in the introduction (lines 54-56) an important but often overlooked aspect towards validating a blood biomarker is to establish that the changes in the blood are reflective of changes in the muscle. This manuscript addresses this issue. There may be potential to use the oxidation of plasma albumin in other models of injury or disease as the reviewer suggests, however, this was not the focus of the submitted manuscript.
We do not agree with the reviewer that different ages and exercise did not contribute much. Researchers use different ages of mdx mice for experimental studies, and exercise to induce myofibre damage. Therefore, we include a range of ages, and an exercise study, to examine the relationship between protein thiol oxidation in muscle tissue and blood.
Reviewer 3 Report
Most of the concerns in terms of adding mice groups were not address but the rest of the concerns were address. If no more mice groups are not added, at least some explanation can be added to manuscript
Author Response
Most of the concerns in terms of adding mice groups were not address but the rest of the concerns were address. If no more mice groups are not added, at least some explanation can be added to manuscript.
As recommended by the reviewer we have modified the manuscript. We direct readers to previous publications (lines 45-50), so that readers are aware that there are publications examining protein thiol oxidation in different animal models, muscle and experimental conditions.
With respect to the age groupings we now include the following statement (lines 104-106) to alert the reader that not all age groups used experimentally were examined:
“These ages were chosen on the basis of changes in muscle pathology. The ages also include many (but not all) ages that are routinely used experimentally”.
We have also included a statement (lines 388-389) to explain the purpose of the exercise study:
“In adult mdx mice, exercise is used to increase dystrophic myofibre damage [51-54], and for this reason we included a study to examine the impact of exercise on albumin thiol oxidation”.